# Targeted Delivery of Antifungal Liposomes to *Rhizopus delemar*

**DOI:** 10.3390/jof8040352

**Published:** 2022-03-30

**Authors:** Quanita J. Choudhury, Suresh Ambati, Zachary A. Lewis, Richard B. Meagher

**Affiliations:** 1Department of Microbiology, University of Georgia, Athens, GA 30602, USA; quanita.choudhury@uga.edu (Q.J.C.); zlewis@uga.edu (Z.A.L.); 2Department of Genetics, University of Georgia, Athens, GA 30602, USA; asuresh@uga.edu

**Keywords:** mucormycosis, *Rhizopus delemar*, C-type lectin receptors, Dectin-1, amphotericin B, liposomes, DectiSomes, oligoglucans, beta-glucan

## Abstract

Mucormycosis (a.k.a. zygomycosis) is an often-life-threatening disease caused by fungi from the ancient fungal division Mucoromycota. Globally, there are nearly a million people with the disease. *Rhizopus* spp., and *R. delemar* (*R. oryzae, R. arrhizus*) in particular, are responsible for most of the diagnosed cases. Pulmonary, rhino-orbito-cerebral, and invasive mucormycosis are most effectively treated with amphotericin B (AmB) and particularly with liposomal formulations (e.g., AmBisome^®^). However, even after antifungal therapy, there is still a 50% mortality rate. Hence, there is a critical need to improve therapeutics for mucormycosis. Targeting AmB-loaded liposomes (AmB-LLs) with the pathogen receptor Dectin-1 (DEC1-AmB-LLs) to the beta-glucans expressed on the surface of *Aspergillus fumigatus* and *Candida albicans* lowers the effective dose required to kill cells relative to untargeted AmB-LLs. Because Dectin-1 is an immune receptor for *R. delemar* infections and may bind it directly, we explored the Dectin-1-mediated delivery of liposomal AmB to *R. delemar*. DEC1-AmB-LLs bound 100- to 1000-fold more efficiently to the exopolysaccharide matrix of *R. delemar* germlings and mature hyphae relative to AmB-LLs. DEC1-AmB-LLs delivering sub-micromolar concentrations of AmB were an order of magnitude more efficient at inhibiting and/or killing *R. delemar* than AmB-LLs. Targeted antifungal drug-loaded liposomes have the potential to improve the treatment of mucormycosis.

## 1. Introduction

Globally, there are approximately 900,000 individuals with mucormycosis, mostly in India [1,2]. Among those at particular risk are patients with lung diseases; neutropenic patients, such as those receiving prolonged immunosuppression for hematopoietic stem cell transplants; patients receiving long-term treatment for inflammatory diseases; and patients with diabetic ketoacidosis, COVID-19, or AIDS [3,4,5,6,7,8,9,10]. The number of reported cases of mucormycosis has increased 6- to 7-fold in the last four decades [7], paralleling the increasing numbers of individuals on immunosuppressants and very recently COVID-19. Among the diverse Mucoromycota [11], the genus *Rhizopus* and one species in particular, *Rhizopus delemar* (*R. oryzae*, *R. arrhizus*), are responsible for 50% or more of all diagnosed cases [7,12,13]. *R. delemar* is an opportunistic pathogen living in soil on rotting vegetation. The primary infection route is via inhalation of its sporangiospores, which leads most commonly to pulmonary and rhino-orbito-cerebral infections [14]. Liposomal amphotericin B (AmB) followed by isavuconazole (ISZ) and/or posaconazole (POS) are the most commonly prescribed antifungals [15]. The surgical removal of infected tissue prior to antifungal therapy significantly improves the outcome [16,17]. However, even with antifungal therapy and surgery, there is still approximately a 50% to 99% mortality rate within several months of diagnosis depending upon the level of dissemination at the time of accurate diagnosis and treatment [7,14,16,18]. Clearly, there is a critical need for improved antifungal therapies for mucormycosis.

The immune response to infections caused by *Rhizopus* spp. is mediated by signaling from the C-type lectin pathogen receptor Dectin-1 (*CLEC7A*) [12]. Dectin-1 is expressed on the surface of some classes of leukocytes, including dendritic cells and neutrophils. Indirect evidence suggests Dectin-1 may bind directly to oligoglucans expressed by *Rhizopus* [19,20]. Two Dectin-1 monomers float together such that their extracellular carbohydrate recognition domains (CRDs) form homo-dimers that bind with high affinity to beta-glucans in the cell wall and/or the exopolysaccharide matrices of pathogens [21]. We have been developing DectiSomes as anti-infective agents, using C-type lectin pathogen receptors to target liposomes loaded with antifungal drugs to pathogenic fungi [22,23,24]. We have shown that the CRD and stalk region of Dectin-1 may be tethered to liposomes loaded with antifungal drugs, targeting these liposomes specifically to beta-glucans on the surface of fungal pathogens [23,25]. As designed, Dectin-1 CRD monomers float in the liposomal membrane and form the functional homo-dimers necessary for beta-glucan binding. Dectin-1-coated, AmB-loaded liposomes (DEC1-AmB-LLs) bind to the cell walls and exopolysaccharides of *Aspergillus fumigatus* and *Candida albicans* orders of magnitude more strongly than untargeted AmBisome^®^-like AmB-LLs. DEC1-AmB-LLs also inhibit and/or kill in vitro-grown *A. fumigatus* 100-fold more efficiently than AmB-LLs, reducing the in vitro effective dose for 90% killing more than 10-fold. Considering that Dectin-1 might bind directly to *R. delemar*, we explored the binding of DEC1-AmB-LLs to *R. delemar* and their potential to enhance the efficacy of antifungal liposome treatment. 

## 2. Materials and Methods

### 2.1. Fungal Strain and Culture Conditions

All studies were performed with *R. delemar* strain 99–880 (ATCC MYA-4621). Sporangiospore stocks were stored frozen at −80 °C in 20% glycerol and were prepared fresh for the following experiments by harvesting sporangiospores from potato dextrose agar (PDA; ThermoFisher, Cat# 0013-01-4, Waltham, MA, USA) plates. PDA plates were inoculated with 2 × 10^6^
*R. delemar* sporangiospores, which were evenly spread with sterile glass beads and incubated at 37 °C. After three days, 1× phosphate-buffered saline (PBS) + 0.05% tween was added to the surface of the plates, and sporangiospores were scraped from the surface into a sterile 250 mL beaker. The sporangiospores were then filtered through a 40 µm cell strainer (Fisherbrand, Cat# 22363547, Rockingham County, NH, USA) into a 50 mL conical tube. The tube was centrifuged at room temperature for five mins at 1200× *g*. The supernatant was removed, and the sporangiospore pellet was resuspended in either 1 mL of PBS + 0.05% tween for short-term storage at 4 °C or in 1 mL of 20% glycerol in sterile deionized water for long-term storage at −80 °C. Titers were determined via hemocytometer counts. Sporangiospores stored at 4 °C remained 99% viable for one to two months. 

### 2.2. Liposome Preparations and Fluorescent Tagging of Dectin-1

We prepared 100 nanometer (nm)-diameter pegylated AmB-LLs that contained 11 mole % AmB and 2 mole % rhodamine relative to moles of liposomal lipid as previously described [25]. Pegylation extends the half-life of packaged drugs by significantly reducing opsonization and phagocytosis [26,27]. We previously showed that our pegylated AmB-LLs significantly outperformed commercial AmBisome^®^ at reducing fungal burden in a mouse model of candidiasis [23], presumably because of pegylation. The AmB-LLs were then coated with either 1 mole % Dectin-1 or 0.33 mole % bovine serum albumin (BSA; Sigma-Aldrich, Cat# A-8022, St. Louis, MO, USA), also as described, which achieves the same microgram protein concentration on the surface of the two liposome preparations [25]. Liposomes were stored at 4 °C in RN#5 buffer (0.1 M NaH_2_PO_4_, 10 mM triethanolamine pH 8.0, 1 M L-arginine, 100 mM NaCl, 5 mM EDTA, and fresh 5 mM 2-mercaptoethanol) [25], and all preparations were adjusted to contain 600 μM AmB. Each liposome preparation was freshly reduced with 1 mM 2-mercaptoethanol (BME; Sigma-Aldrich, Cat# M7522, St. Louis, MO, USA) on a monthly basis and again just prior to use. Rhodamine B-conjugated Dectin-1 protein, DEC1-Rhod, was prepared following the protocol we described previously for rhodamine-tagging Dectin-2 [28]. 

### 2.3. Liposome Binding Activity

In order to prepare fixed agar plugs for microscopy (Appendix A), *R. delemar* sporangiospores were plated on 1.5% agar plates made with RPMI-1640 media lacking phenol red dye (Sigma-Aldrich, Cat# R8755, St. Louis, MO, USA) + 0.165 M MOPS (3-(N-morpholino)propanesulfonic acid) (Sigma-Aldrich, Cat# M1254, St. Louis, MO, USA) [29] adjusted to pH 7 and incubated at 37 °C for either 6 to 8 h to sample swollen sporangiospores and germlings or overnight until the hyphal colonies were approximately 6 cm in diameter. A sterile 7 mm-diameter cork-borer was used to remove plugs from the plate with germlings or from the periphery of hyphal colonies to obtain elongating hyphae. Plugs were deposited into a 24-well plate containing 1 mL of 1× PBS and washed once. Plugs were fixed for one hour in freshly-prepared 3.7% formaldehyde (J.T. Baker, Cat# 2106-01, Phillipsburg, NJ, USA) and washed 3 times in PBS before being stored overnight at 4 °C. Plugs were blocked in liposome dilution buffer 1 (LDB1; 1× PBS + 0.5% BSA + 1 mM BME) for one hour. Dectin-1- and BSA-coated liposomes were diluted to 1:100 *w*/*v* (protein/buffer). AmB-LLs were similarly diluted. Plugs were incubated with liposomes for two hours at room temperature with shaking at 50 rpm and washed once with the LDB1 buffer in the dark. A 25 mM stock of calcofluor white (CW; Bayer Corp., Blankophor BBH, CAS 4193-55-9, Pittsburgh, PA, USA) stored in dimethyl sulfoxide (DMSO; Sigma-Aldrich, Cat# D8418, St. Louis, MO, USA) was diluted 1:5000 in 1× PBS + 5% DMSO, added to the wash, and incubated with the agar plugs for 30 min. After one more wash, plugs were placed on a microscope slide with the hyphal colony facing upward, fitted with a cover slip, and imaged under epifluorescence on a Leica DM6000 compound microscope at 5× or at 63× under oil immersion [25]. CW cell staining was imaged in the DAPI channel (Ex360/Em470) and rhodamine B-tagged liposomes in the Texas red channel (Ex560/Em645). The area of red fluorescent liposome binding was quantified using ImageJ (imagej.nih.gov/ij, v. 1.53a; accessed on 4 May 2020). The following sequence of commands was used: Image > (8 bit), followed by Image > Adjust > Threshold > Apply. The commands Analyze > Measure were used to place the area data for each image in a file in Excel (v. 16.16.27). Live hyphal plugs were prepared by omitting the formaldehyde and subsequent wash steps.

In order to assess the specificity of DEC1-AmB-LL binding, fixed hyphal plugs were prepared as described above. DEC1-AmB-LLs were preincubated with either 0.5 mg/mL of laminarin (Sigma-Aldrich; Cat# L-9634, St. Louis, MO, USA) or 0.5 mg/mL yeast mannans (Sigma-Aldrich; Cat# M3640, St. Louis, MO, USA) for 30 min at room temperature. These liposomes were added to the hyphal plugs and incubated for two hours at room temperature with 50 rpm shaking in the dark. The plugs were then processed and imaged as described above. 

### 2.4. Liposome Inhibition Activity

In order to determine the inhibitory concentration of AmB delivered by DEC1-AmB-LLs relative to AmB-LLs, two variations of the following assay were performed. Eight hundred *R. delemar* sporangiospores were plated in 90 uL of liquid RPMI + 0.165 M MOPS (pH 7) in individual wells of 96-well microtiter plates. Ten uL of each liposome type (AmB-LLs, BSA-AmB-LLs, and DEC1-AmB-LLs) delivering different concentrations of AmB were immediately added to the respective wells, and the plates were incubated for 24 or 48 h at 37 °C with 120 rpm shaking. In one variation, after the first 24 h, the center of each well was imaged with an EVOS imaging system (AMG F1) at 10× magnification. After 48 h, an overview picture of the entire 96-well plate was taken. In the second variation, the cell density (OD at A610) and metabolic activity were measured at 24 h using a Bio-Tek Synergy HT fluorescent microtiter plate reader. Twenty microliters of the CellTiter-Blue (CTB, Promega; Cat.# G8080; Madison, WI, USA) resazurin reagent was added to each well according to the manufacturer’s instructions. The plate was incubated at 37 °C for two hours. The pink fluorescence in each well was quantified (Ex485/Em590). The background fluorescence in wells lacking cells was subtracted.

To determine metabolic activity following a short exposure to targeted and untargeted AmB-LLs, 5000 *R. delemar* sporangiospores were plated in 90 uL of liquid RPMI + 0.165 M MOPS (pH 7) in a 96-well microtiter plate. The plate was incubated at 37 °C for eight hours until uniform germination and mature hyphae were observed. The liposome treatments were added as described above, and the plate was incubated for three hours at 37 °C. The CTB assay was then performed as described above. 

### 2.5. Data Management

Quantitative imaging data from ImageJ were initially managed in Excel. Imaging data were then moved to Graph Pad Prism 9 (v. 9.3.1), where scatter bar plots were prepared and standard errors were estimated. *p* values were estimated in Excel using the Student’s two-tailed *t* test, T.TEST, for various comparisons.

## 3. Results

### 3.1. Dectin-1 Targeted DEC1-AmB-LLs Bind to R. delemar

To determine if Dectin-1 would target liposomes to *R. delemar*, we grew *R. delemar* on the surface of agar plates and stained the cells with rhodamine B-tagged liposomes. Liposome staining was viewed top down by epifluorescence. Figure 1 shows the binding of BSA-AmB-LLs, AmB-LLs, or DEC1-AmB-LLs to early stages of germinating *R. delemar* sporangiospores. Cells are visible due to the fluorescent stain CW. DEC1-AmB-LLs bound efficiently to the exopolysaccharide matrix and, to a lesser extent, the cell wall of swollen and germinating sporangiospores (Figure 1C), while BSA-AmB-LLs and AmB-LLs did not bind efficiently (Figure 1A,B). DEC1-AmB-LLs also bound efficiently to the exopolysaccharide matrix of germlings (Figure 1F), while BSA-AmB-LLs and AmB-LLs did not bind efficiently (Figure 1D,E). By measuring the area of red fluorescent liposome binding to randomly photographed fields of CW-stained germlings, we quantified the binding efficiency. DEC1-AmB-LLs bound 126-fold (*p* = 7.4 × 10^−4^) more efficiently than AmB-LLs (Figure 1G). The fact that essentially all germlings efficiently bound to DEC1-AmB-LLs is made evident by examining the distribution of data in the scatter bar plot. 

Figure 2 examines liposome binding to hyphae grown on agar. DEC1-AmB-LLs (Figure 2C) bound efficiently to exopolysaccharide distributed all along the majority of the hyphae, while control liposome binding was barely detectable (Figure 2A,B). Figure 2D quantifies these data, showing that DEC1-AmB-LLs bound approximately 1900-fold (*p* = 1.3 × 10^−10^) more strongly than AmB-LLs. At 63× magnification (Figure 2E) we observed that DEC1-AmB-LLs bound along the cell wall (see arrows) and even more strongly to extracellular deposits of exopolysaccharide. We could not confirm if binding that appeared to be on the cell wall was to the cell wall itself or to small deposits of exopolysaccharide on the wall surface. We performed a parallel binding experiment on live hyphae (Figure 2F) and observed that DEC1-AmB-LLs bound 368-fold more strongly than AmB-LLs (*p* = 0.002). It is likely that the water solubility of some exopolysaccharides in unfixed cell preparations removed a portion of their beta-glucans, accounting for the slightly lower level of binding relative to that observed for fixed cells [30]. The remodeling of the exopolysaccharide matrix in live cells could also account for some loss of binding. Complete biological replicates gave similar results for both fixed and live cells (Appendix A). It is worth noting that coating AmB-LLs with BSA (BSA-AmB-LLs) did not significantly alter binding (Figure 1G and Figure 2D,F). Hence, there does not appear to be a significant non-specific affinity of protein-coated liposomes for *Rhizopus.*

It seemed possible that the 100 nm-diameter size of our DEC1-AmB-LLs limited penetration and binding to cell wall beta-glucans. Therefore, we labeled hyphae with rhodamine-conjugated Dectin-1 protein. Dectin-1 is projected to have a diameter measured in tens of angstroms [21]. We labeled fixed hyphae with DEC1-Rhod. The exopolysaccharide staining pattern was indistinguishable from that of DEC1-AmB-LLs (Appendix A).

The glycan specificity of Dectin-1-targeted liposome binding to *R. delemar* was evaluated by a competitive inhibition study with laminarin (6 kDa), which contains cognate beta-glucan ligands, and yeast mannan (133 kDa), which is composed of various oligomannans (Figure 3). Dectin-1 binds to various beta-glucan crosslink variants with dissociation constants (e.g., Kds) ranging from mM to pM [31]. Laminarin is expected to contain many, but certainly not all, of the variously crosslinked oligoglucan structures found among fungal polysaccharides. DEC1-AmB-LLs were preincubated with laminarin or yeast mannan before being bound to mature *R. delemar* hyphae. Relative to the DEC1-AmB-LL untreated control (Figure 3A), laminarin inhibited DEC1-AmB-LL binding (Figure 3C), whereas yeast mannan did not (Figure 3B). We quantified the area of red fluorescent liposome binding from multiple images. Laminarin inhibited DEC1-AmB-LL binding 4.2-fold (*p* = 4.7 × 10^−8^) relative to the DEC1-AmB-LL untreated control (Figure 3D). A biological replicate of this experiment is shown in Appendix A.

### 3.2. DEC1-AmB-LLs Efficiently Inhibit and/or Kill R. delemar

Multiple assays were used to examine the ability of liposomal AmB to reduce the viability and/or growth of *R. delemar*. First, we inoculated 96-well microtiter plates with *R. delemar* sporangiospores and immediately added BSA-AmB-LLs, AmB-LLs, and DEC1-AmB-LLs, delivering final concentrations of AmB ranging from 6.4 μM in a two-fold dilution series down to 0.0125 μM (Figure 4). Visual inspection of the plate revealed that the minimum inhibitory concentration (MIC) after 48 h of growth for DEC1-AmB-LLs was 0.4 μM, while that for AmB-LLs and BSA-AmB-LLs was 3.2 μM, an 8-fold difference (Figure 4A). It was possible to take a closer look at each well after only 24 h of growth, when the hyphae were less densely packed; images taken at 10× magnification revealed a similar result (Figure 4B). DEC1-AmB-LLs delivering 0.4 μM AmB effectively inhibited growth, while AmB-LLs and BSA-AmB-LLs delivering 1.6 and 6.4 μM, respectively, inhibited growth. A complete biological replicate of this experiment, shown in Appendix A, suggests a similar reduction in the MIC by DEC1-AmB-LLs relative to AmB-LLs.

Secondly, using the same regimen for growing and treating cells as in Figure 4, we quantified cell density and metabolic activity (Figure 5). We found that DEC1-AmB-LLs delivering a range of concentration from 0.2 μM to 1.6 μM AmB were significantly more effective at reducing cell density and metabolic activity than AmB-LLs. For example, at 0.4 μM, DEC1-AmB-LLs reduced cell density 23.6-fold (*p* = 5.8 × 10^−4^) (Figure 5A) and metabolic activity 75-fold (*p* = 3.9 × 10^−12^) (Figure 5B) relative to AmB-LLs. CTB is a metabolic activity assay that measures the reduction of resazurin to fluorescent resorufin, which is dependent upon an intact electron transport chain in living cells. Preliminary experiments using this CTB assay design had shown improved dose-dependent inhibition and killing activity for DEC1-AmB-LLs in this range of AmB concentrations relative to AmB-LLs (Appendix A). Replicates of these experiments are shown in Appendix A. 

Third, we wished to determine how rapidly targeted liposomes had an impact on metabolic activity. Sporangiospores were germinated to early hyphal stage, treated for three hours with BSA-AmB-LLs, AmB-LLs, and DEC1-AmB-LLs delivering 1.56 μM, 3.12 μM, and 6.25 μM AmB, and assayed with CTB reagent (Figure 5C). At 3.12 and 6.25 μM AmB, DEC1-AmB-LLs were 1.9-fold (*p* = 0.003) and 3.2-fold (*p* = 1.5 × 10^−4^) more effective at reducing metabolic activity than AmB-LLs, respectively. A biological replicate gave a similar result (Appendix A).

## 4. Discussion

Dectin-1 recognizes beta-glucans that are present in fungal cell walls and exopolysaccharide matrices but are sometimes masked by other molecular components. We showed that Dectin-1 was extremely efficient at targeting AmB-loaded liposomes, DEC1-AmB-LLs, to *R. delemar* swollen sporangiospores, germlings, and mature hyphae. We observed DEC1-AmB-LLs bound primarily to the exopolysaccharide matrix and less so to the cell wall or to exopolysaccharide deposited close to the cell wall. Rhodamine-tagged Dectin-1 protein bound with the same specificity to *R. delemar’s* exopolysaccharide. Hence, it appears that the 100 nm-size of DEC1-AmB-LLs did not significantly limit liposome access to its cognate ligands. DEC1-AmB-LLs were significantly and dramatically more effective at inhibiting and/or killing *Rhizopus* in vitro than untargeted AmB-LLs or BSA-AmB-LLs. Using both cell growth and metabolic assays, we observed that DEC1-AmB-LLs delivering sub-micromolar concentrations of AmB were significantly more efficient at inhibiting and/or killing *R. delemar* than untargeted AmB-LLs. We were able to detect significant loss of metabolic activity within three hours of treatment. 

AmB has several partially validated antifungal activities related to its affinity for ergosterol (Erg) in the fungal bilipid membrane, including opening ion channels in the membrane to cause lethal ion leakage and extracting Erg from the lipid bilayer to the membrane surface, which also compromises the membrane [32]. Our results do not distinguish among the various mechanisms of AmB’s activity. Yet, our data robustly demonstrate that that Dectin-1-targeted DEC1-AmB-LLs were more efficiently associated with *R. delemar*’s exopolysaccharides and had greater antifungal activity than either AmB delivered in AmB-LLs or our protein-coated control BSA-AmB-LLs. Therefore, it does not appear that AmB itself plays a measurable role in the enhanced efficacy of targeted liposomes. 

Each DEC1-AmB-LL DectiSome contains several thousand molecules of rhodamine B that enhance signal intensity and more than a thousand Dectin-1 receptor molecules on its surface, enabling multimer formation that enhances the avidity of binding to cognate oligoglycans [25]. If a C-type lectin receptor protein was used alone in a fungal cell binding study and assayed by immunofluorescence, signal intensities might be reduced by orders of magnitude relative to that achieved by a fluorescent DectiSome. This makes DectiSomes excellent reagents for examining the direct binding of different C-type lectins to various pathogens [22,23,24,25,28]. 

The Mucoromycota is an ancient division of the fungal kingdom that contains a large number of morphologically diverse human pathogens that cause mucormycosis [33,34]. They are estimated to have diverged from a common ancestor in the fungal tree of life nearly 1.3 billion years ago [35,36]. Hence, it is not surprising that the glycan composition of the Mucoromycota cell wall and exopolysaccharide matrix [19,37,38,39] appear to be distinct from other pathogenic fungi [30,40,41,42]. The sporangiospore and hyphal cell wall [38] and the exopolysaccharide matrix [39] each are composed of approximately 43% glucose; other components include lower amounts of N-acetyl-glucosamine, mannose, fructose, lipids, proteins, and phosphate [38,39]. Considering that Dectin-1 recognizes oligo-beta-glucans, it is not surprising that Dectin-1-targeted liposomes bound to *Rhizopus*. The weaker binding we observed to the cell wall relative to the exopolysaccharide of *Rhizopus* suggests that most of the cell wall oligoglucans were masked from DEC1-AmB-LL binding. Experiments with DectiSomes targeted by the oligo-mannan-specific C-type lectin Dectin-2 are ongoing. 

While liposomal AmB formulations such as AmBisome^®^ delivering as much as 10 mg/kg/day are significantly less toxic than alternative AmB therapies, the several-months-long therapies needed to clear mucormycosis still result in infusion-related reactions and nephrotoxicity [43,44,45]. If DEC1-AmB-LLs can reduce the effective dose of liposomal AmB and/or reduce the duration of treatment in the clinic, this should reduce the risk of patients developing toxic effects from AmB. Salvage therapies after patients become intolerant to AmB include very high doses of posaconazole (POS) or isavuconazole (ISZ) on the order of hundreds of mg/kg/day [46,47,48,49,50]. Even if Dectin-1-targeted liposomes improve the performance of POS or ISZ by 10-fold in the clinic, it may not be cost-effective to prepare targeted liposomes that deliver tens of mg/kg/day doses of these drugs. However, our data also suggest that DEC1-AmB-LLs kill *Rhizopus* faster than untargeted therapies. Enhanced speed of killing may enable patients to clear *Rhizopus* infections with drug regimens of shorter duration or with fewer treatments, reducing the risk of AmB toxicity. 

Immunoliposomes have been used in the clinic for some time to target anti-cancer drugs to cancer cells and tumors. They generally improve drug efficacy several-fold over untargeted drugs [51,52,53]. Although conceptually DectiSomes function similarly to immunoliposomes by targeting drugs to pathogenic cells, DectiSomes have some distinct advantages [24]. C-type lectin receptors such as Dectin-1 generally recognize a much wider variety of target ligands than monoclonal antibodies, which supports their development as pan-antifungal reagents. Dectin-1 in particular recognizes the beta-glucans produced by nineteen of the twenty genera of pathogenic fungi [12]. Once developed for one fungal pathogen in the clinic, it should not be difficult to broaden their application to other pathogens. In addition, C-type lectins are much less expensive to produce than monoclonal antibodies, which will favor their development as reagents to treat fungal diseases in less wealthy countries [54,55,56,57]. Finally, low production costs may encourage the pharmaceutical industry to expend the large amounts of capital needed to develop DectiSomes. 

In conclusion, there is a pressing demand for more effective therapeutics to treat mucormycosis because even after surgery and drug treatment, there is still a high mortality rate. We have shown order of magnitude improvements in the in vitro performance of AmB against *R. delemar* when delivered by Dectin-1-targeted liposomes. It appears that targeting liposomal AmB to the exopolysaccharide matrix of *Rhizopus* is sufficient to significantly improve liposomal drug performance. Future experiments will focus on mouse models of mucormycosis, including determining if Dectin-1-targeted liposomes bind to *R. delemar* at infection sites in the lung, reduce fungal burden in the lungs, and improve mouse survival. We also will need to confirm that Dectin-1-targeted liposomes work effectively against other clinically relevant members of Mucoromycota [58] in light of their ancient diversity [34].

## Figures and Tables

**Figure 1 jof-08-00352-f001:**
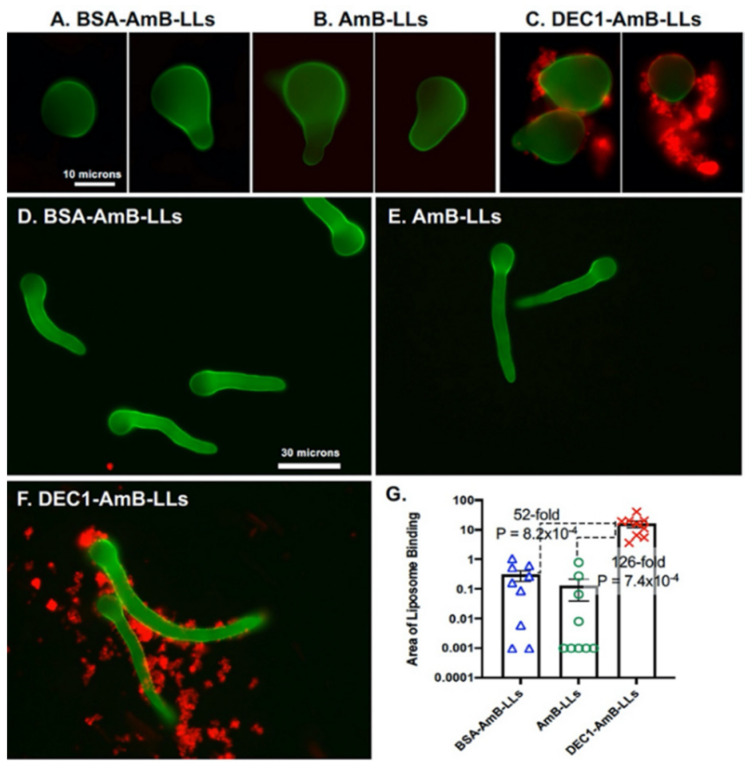
Dectin-1-targeted AmB-loaded liposomes, DEC1-AmB-LLs, bind efficiently to germinating sporangiospores and germlings of *R. delemar*. (**A**–**C**) Representative fluorescence images of swollen *R. delemar* sporangiospores (63× magnification) on an agar surface are shown. Sporangiospores were either treated with BSA-AmB-LLs (**A**), AmB-LLs (**B**), or DEC1-AmB-LLs (**C**). (**D**–**F**) Representative fluorescence images of *R. delemar* germlings (63× magnification) are shown. Germlings were either treated with BSA-AmB-LLs (**D**), AmB-LLs (**E**), or DEC1-AmB-LLs (**F**). (**G**) The area of red liposome binding (log_10_) was quantified as shown (N = 9). Standard errors and *p*-values are included. Size bars indicate 10 and 30 micron scales.

**Figure 2 jof-08-00352-f002:**
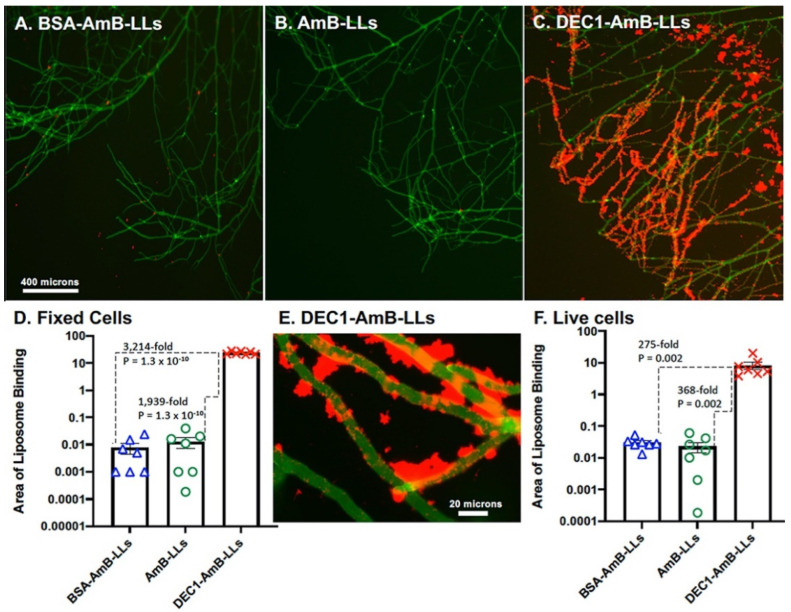
Dectin-1-targeted liposomes bind efficiently to *R. delemar* hyphae. (**A**–**C**) Representative fluorescence images of *R. delemar* hyphae (5× magnification) are shown. Hyphae grown on an agar surface were fixed and either treated with BSA-AmB-LLs (**A**), AmB-LLs (**B**), or DEC1-AmB-LLs (**C**). (**D**) The area of red liposome binding (log_10_) for fixed cells was quantified as shown (N = 7). (**E**) A fluorescence image of DEC1-AmB-LLs binding to fixed *R. delemar* hyphae at 63× magnification is shown. (**F**) The area of red liposome binding (log_10_) for live hyphae was quantified as shown (N = 7). Standard errors and *p*-values are included. Size bars indicate 400 and 20 micron scales for the 5× and 63× images, respectively.

**Figure 3 jof-08-00352-f003:**
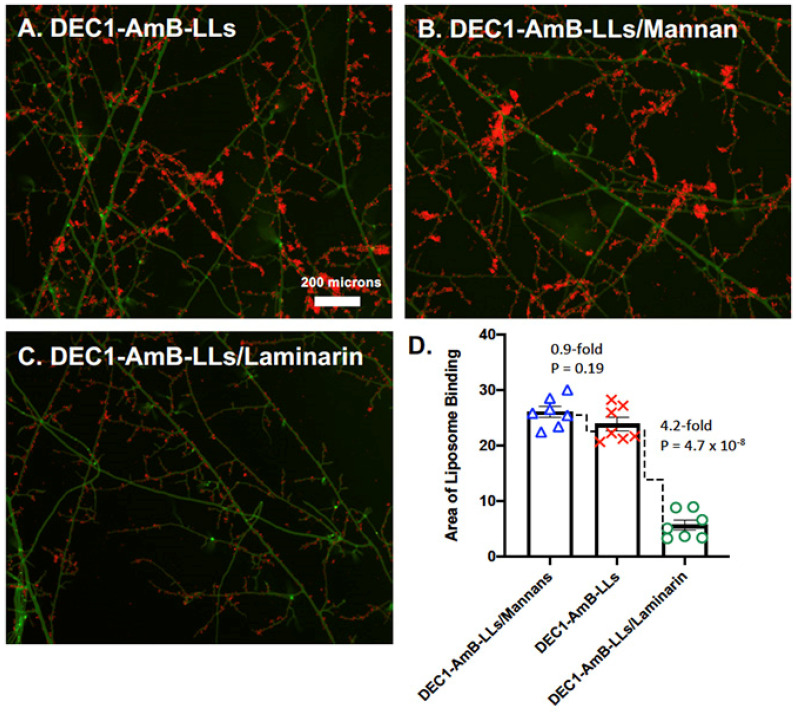
DEC1-AmB-LL binding to *R. delemar* is beta-glucan-specific. (**A**–**C**) Representative fluorescence images of *R. delemar* hyphae (5× magnification) on an agar surface are shown. Hyphae were treated with DEC1-AmB-LLs that had been preincubated either with control buffer (**A**), yeast mannans (**B**), or the beta-glucan laminarin (**C**). The size bar indicates 200 microns. (**D**) The area of red liposome binding (linear plot) was quantified (N = 7). Standard errors and *p*-values are included.

**Figure 4 jof-08-00352-f004:**
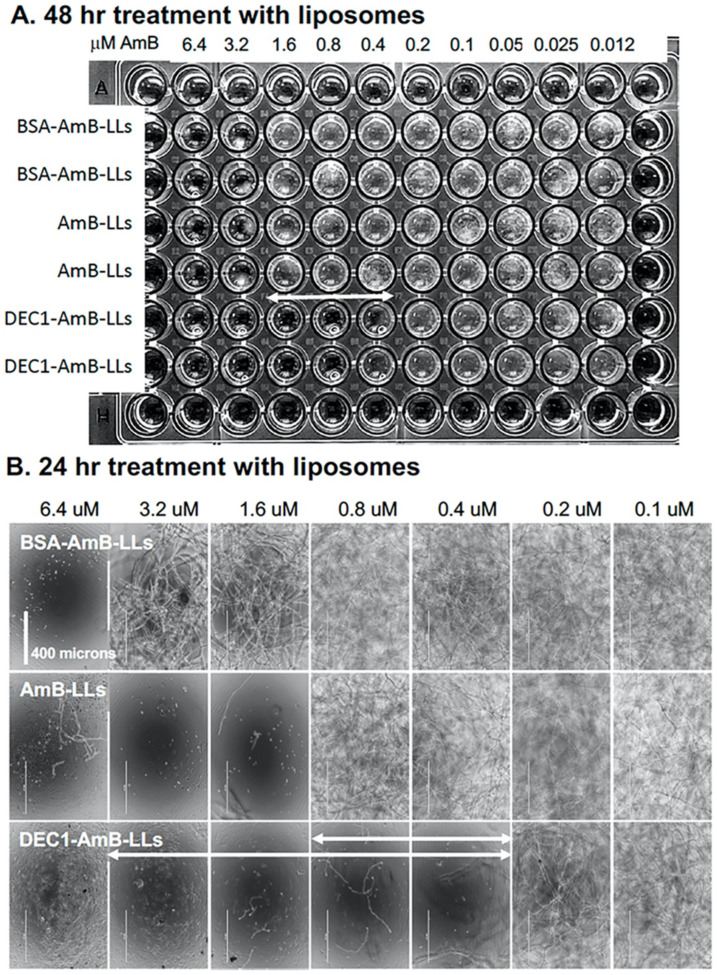
Qualitative DEC1-AmB-LL inhibition and killing assays based on cell growth and density. (**A**) *R. delemar* sporangiospores diluted into liquid RPMI + MOPS media were immediately treated with either BSA-AmB-LLs, AmB-LLs, or DEC1-AmB-LLs delivering the indicated AmB concentrations and incubated at 37 °C. An overview image showing hyphal density was taken 48 h after treatment. The two-headed arrow indicates the range at which DEC1-AmB-LLs significantly inhibited *R. delemar* growth as compared to the control liposome treatments. (**B**) Images of the center of each well from the same plate had been taken 24 h after treatment at 10× magnification. The two two-headed arrows indicate the ranges at which DEC1-AmB-LLs significantly inhibited *R. delemar* growth compared to the controls at this earlier time point. The size bar indicates 400 microns.

**Figure 5 jof-08-00352-f005:**
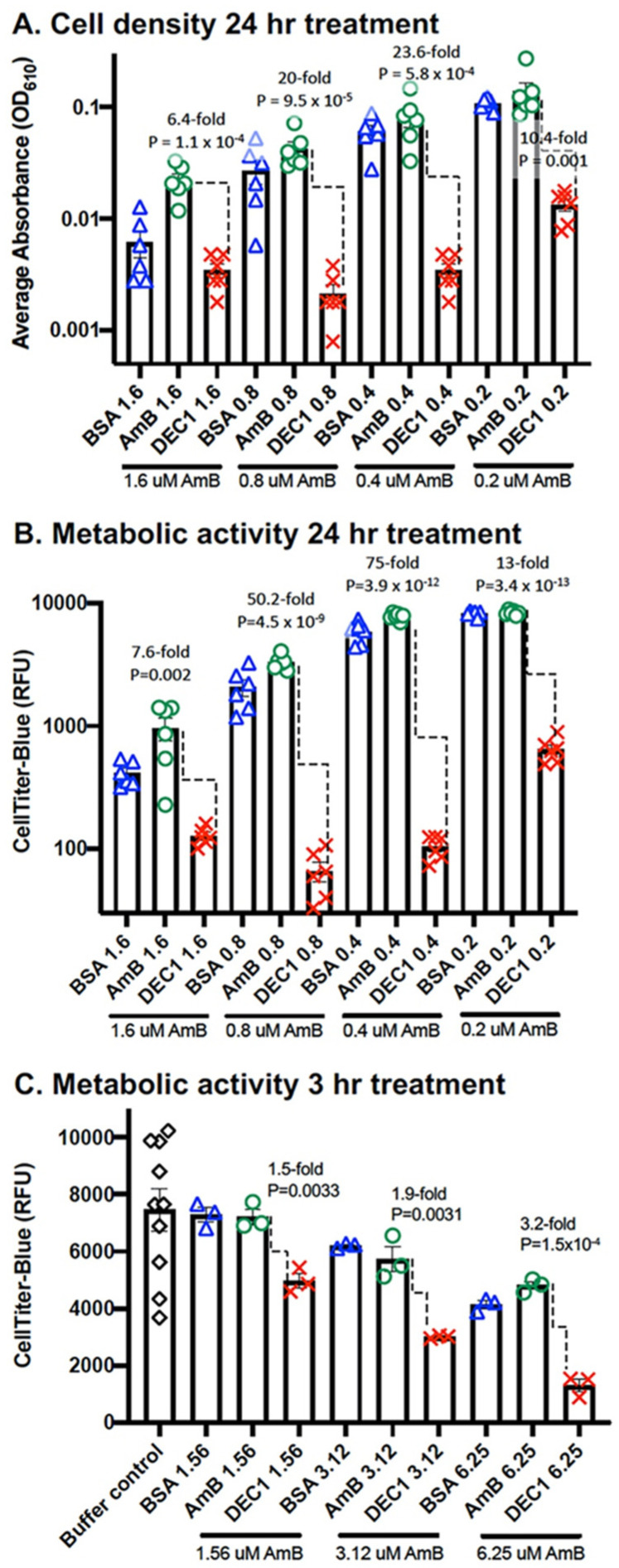
Quantitative DEC1-AmB-LL inhibition and killing assay based on cell density and metabolic activity. (**A**) Microtiter plate-grown cells were treated with liposomes delivering the indicated concentrations of AmB for 24 h and cell density was quantified. (**B**) The same 24 h microtiter plate-grown cells were incubated with CTB reagent and metabolic activity was quantified. (**C**) *R. delemar* sporangiospores were diluted into liquid RPMI + MOPS media and incubated at 37 °C for 8 h. *R. delemar* hyphae were then treated for 3 h with either BSA-AmB-LLs, AmB-LLs, or DEC1-AmB-LLs at the noted AmB concentrations or treated with buffer alone (buffer control). The residual metabolic activity was quantified as relative fluorescence units (RFU) generated from the electrochemical reduction of the CellTiter-Blue reagent to a fluorescent product (N = 3). Standard errors, fold differences, and *p*-values are indicated.

## Data Availability

All new data that are discussed are presented within this publication and its data supplement, and any data obtained from other publications were appropriately cited.

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
