# Peer review of "Targeted Delivery of Antifungal Liposomes to Rhizopus delemar"

_jof, 2022, doi:10.3390/jof8040352_

Round 1

Reviewer 1 Report

This is an excellent manuscript that describes an important advance in the treatment of a disease for which novel therapies are desperately needed.

The only thing missing that I would have liked to see is how well the Dectisomes killed other species of Mucorales. However, given that the authors focused on the most commonly isolated species and they have previously demonstrated efficacy on very divergent fungi, there is an extremely high likelihood that these Dectisomes would work on other Mucorales.

This manuscript is very clearly written.

Congratulations on the excellent work.

Author Response

We are submitting our revised manuscript for the special issue of JOF, which is focused on mucormycosis and aspergillosis. We have responded to all the reviewers’ comments and suggestions with modifications to the text and the addition of one experiment mentioned in the text and its results shown in Supplemental Fig. SF2C. Eight new references were added to meet the reviewers’ concerns.

Reviewer 1. This reviewer had a minor concern as to if this method would kill “other species of Mucorales.”  

Line 388. We added the following. “We also will need to confirm that Dectin-1 targeted liposomes work effectively against other clinically relevant members of Mucoromycota in light of their ancient diversity.” 

Reviewer 2 Report

The article bt Choudhury et al, describes a new modification of Amphotericin B. The modification helps the antifungal to significantly bind to beta glucan. Further, the modified Amphoterin B is more effective than standard amphotericin B in inhibiting mucor growth. The article is well written and is an interesting read. Here are my comments:

1) Since Dec1-AmB-LLs bind to beta glucan efficiently does the binding affects the beta-glucan content? If so the authors can do a simple experiment as described below to quantify the beta glucan content in response to the three type of AmB.

https://pubmed.ncbi.nlm.nih.gov/35164553/

If this is true then the Dec1-AmB-LL can be very important in using in combination with caspofungin. Since polyene -caspofungin combination are used in treating mucormycosis (https://pubmed.ncbi.nlm.nih.gov/31667058/)  as well there by lower the cytotoxicity even further.

2) Since amphotericin B targets ergosterol, ergosterol quantification is useful in this case using all three types of AmB. Please clarify.

3) How cytotoxic is DEC1-AmB-LL? In other words how are the authors sure that using DEC-AmB-LL will actually lower the toxicity levels even using at lower concentrations?

4) How stable is the DEC1-AmB-LL in vivo?

Author Response

We are submitting our revised manuscript for the special issue of JOF, which is focused on mucormycosis and aspergillosis. We have responded to all the reviewers’ comments and suggestions with modifications to the text and the addition of one experiment mentioned in the text and its results shown in Supplemental Fig. SF2C. Eight new references were added to meet the reviewers’ concerns.

Reviewer #2. Major concerns

  1. The reviewer suggested we quantify the effect of DEC1-AmB-LL binding on beta-glucan content as per a “simple experiment” as described in Silva et al., mBio, 2022. If such an effect were observed, this would suggest combination therapy with caspofungin was warranted. In the Silva et al paper the beta-glucan content (Fig. 3C) was quantified with aniline blue. However, aniline blue was previously reported not to bind to oryzae (Nichols et al., Mycological Research, 1994). So, it is doubtful this simple method will work for us. Considering the solid quantitative data we have for the antifungal activity of DEC1-AmB-LLs, it might be more direct for us to assay the impact of co-treating with caspofungin now that we have good assays for measuring that effect. In any case, we agree co-treatments with caspofungin may impact DEC1-AmB-LL’s antifungal efficiency. Finally, it is worth noting that fungal beta-glucans represent a wide variety of crosslink glucan variants, and hence, it is not surprising that Dectin-1 binds to these variants with dissociation constants (e.g., Kd) ranging over nine logs (base 10) from mM to pM (Adams, J Pharmacol Exp Ther. 2008). This heterogeneity of composition and binding complicates various methods of interpreting changes in total beta-glucan content. This heterogeneity may account for the lack of aniline blue binding to Rhizopus.

We had partially addressed this issue in our original manuscript by stating the following: “Laminarin is expected to contain many, but certainly not all, of the variously crosslinked oligoglucan structures found among fungal polysaccharides.”

Line 251. To further address the reviewer’s concern, we added the following statement. “Dectin-1 binds to various beta-glucan crosslink variants with dissociation constants (e.g., Kds) ranging from mM to pM [Adams, 2008 #38418].”

  1. Reviewer #2 states. Since amphotericin B targets ergosterol, ergosterol quantification is useful in this case using all three types of AmB. Please clarify.

Yes, for example, it is possible that DEC1-AmB-LLs extracted more Erg from the fungal cell membrane to the surface of membrane than untargeted AmB-loaded liposomes. The impact of targeting on the various mechanisms of AmB’s antifungal activity, although quite interesting, is beyond the scope of this manuscript. We addressed this concern of Reviewer #2’s and a related concern of Reviewer #3, with the following addition to the revised manuscript.

Line 344. “AmB has several partially validated antifungal activities related to its affinity for ergosterol (Erg) in the fungal bilipid membrane, including opening ion channels in the membrane to cause lethal ion leakage and extracting Erg from the lipid bilayer to the membrane surface, which also compromises the membrane 31. Our results do not distinguish among the various mechanisms of AmB’s activity. Yet, our data robustly demonstrate that that Dectin-1 targeted DEC1-AmB-LLs were more efficiently associated with R. delemar’s exopolysaccharides and had greater antifungal activity than either AmB delivered in AmB-LLs or our protein-coated control BSA-AmB-LLs. Therefore, it does not appear that AmB itself plays a measurable role in the enhanced efficacy of targeted liposomes.”

  1. Reviewer #2 asks. How cytotoxic is DEC1-AmB-LL? In other words, how are the authors sure that using DEC-AmB-LL will actually lower the toxicity levels even using at lower concentrations?

No, without experimental evidence we cannot be sure. But, it would be most surprising if in vivo toxicity was not dependent upon AmB dose and treatment duration. In the clinic it is clear that high doses and long-term treatments with any formulation of AmB contribute to toxicity. In vitro, we showed that AmB toxicity is dose dependent as expected. We have previously shown (Ambati et al, mSphere, 2019) that when DEC1-AmB-LLs deliver the same very high doses of AmB as untargeted liposomal AmB, they are equivalently toxic. In the near future we will address the issues of dose and duration dependent toxicity in mouse models. 

Line 378. The statement we make concerning dose and toxicity is quite cautious. “If DEC1-AmB-LLs can reduce the effective dose of liposomal AmB and/or reduce the duration of treatment in the clinic, this should reduce the risk of patients developing toxic effects from AmB.” 

  1. How stable is the DEC1-AmB-LL in vivo?

We do not have direct in vivo measurements of DEC1-AmB-LL stability. No in vivo experiments are included in this manuscript. We had already stated that our pegylated AmB-loaded liposomes are more effective in vivo in mouse models of candidiasis than their un-pegylated analog AmBisome®, likely because they are pegylated. We expect to be able to confirm that pegylation also extends the half-life of Dectin-1 coated liposomes in mouse models. Here is what our manuscript says about this topic.

Original manuscript statement. “We previously showed that our pegylated AmB-LLs significantly outperformed commercial AmBisome® at reducing fungal burden in a mouse model of candidiasis 22, presumably because of pegylation.”

Line 91. In the revised manuscript we expanded on this topic by adding the following. “Pegylation extends the half-life of packaged drugs by significantly reducing the opsonization and phagocytosis [Ramana, 2015 #37079;Tenchov, 2021 #42219].”

Reviewer 3 Report

Abstract:

I do not find necesary the (a.k.a) however I should include R. oryzae as a synonisms of R. delemar since several reports use the first.

Intro

Line 27: 900,000 cases of mucormycosis? Where? In which time span?

Line 28: Two main risk groups are missing. The most known: poorly controlled diabetics (ketoacidotics) and the newest: COVID-associated (the so-called black fungus).

 M&M:

Line 67: stored frozen at -80°C?

Line 70: please use scientific numeration (2x106)

Line 87:  describe the composition of RN#5

Fig. 1c, 1f and 2c, how was it confirmed that the AMB dec-1 liposomes observed on hyphae, germilings and spores effectively bound to fungal structures by affinity and not by simple chemical reaction? By this I mean how the authors could ensure that such accumulation of liposomes is due to affinity with the receptors and not just to simple chemical forces (lipophilicity for example). This effect was reported by Anderson et al in 2015 (Nat Chem Biol. 2014 May ; 10(5): 400–406). They demostrated that AMB forms extramembranous agregates on lipid bilayers that extract ergosterol from them. This effect that resambes "sponges" (they use this word) kills fungal cells. This effect is produced only by the presence of AMB with no need of DEC1 and/or liposoms. I think that the manuscript would improve if these points are better clarified in the text (not only by a phrase in the discusion line 282 "We provide evidence that Dectin-1 binds directly to Rhizopus").

Author Response

    We are submitting our revised manuscript for the special issue of JOF, which is focused on mucormycosis and aspergillosis. We have responded to all the reviewers’ comments and suggestions with modifications to the text and the addition of one experiment mentioned in the text and its results shown in Supplemental Fig. SF2C. Eight new references were added to meet the reviewers’ concerns.

Review #3.

Minor concerns

Line 9. We added (R. oryzae, R. arrhizus) to the abstract.

Line 27. We added the when and where information about mucormycosis. “900,000 individuals currently with mucormycosis, mostly in India”

Line 31. We added the following. “and patients with diabetic ketoacidosis, COVID-19,”

Line 35. We added “(R. oryzae, R. arrhizus)”

Line 75. We added. “-80oC”

Line 77. 2,000,000 was changed to 2x106

Line 98. The composition of RN#5 buffer was added. (0.1 M NaH2PO4, 10 mM triethanolamine pH 8.0, 1 M L-arginine, 100 mM NaCl, 5 mM EDTA, and fresh 5 mM 2-mercaptoethanol)24

Reviewer #3, Major concerns.

This reviewer ask how we knew that the interaction of our Dectin-1 targeted DectiSomes was due to Dectin-1 binding and not some chemical interaction of amphotericin B (AmB) in our liposomes with the liposome membrane. Considering that our control liposomes also contained the same amount of AmB but did not bind, we believe we have already controlled for such an interaction. However, we have responded in full to this concern with an additional experiment showing the same binding pattern for Dectin-1 protein free of liposomes and some new discussion covering the reviewer’s viewpoint.

Line 88. We changed the section heading from “2.2. Liposome Preparations” to “2.2. Liposome Preparations and Fluorescent Tagging of Dectin-1”

Line 89. We added the following. ..”100 nanometer (nm) diameter”…

Line 98. We included the composition of RN#5 buffer.

Line 102. We added the following. “Rhodamine B conjugated Dectin-1 protein, DEC1-Rhod, was prepared following the protocol we described previously for tagging Dectin-2 [Ambati, 2019 #40694].”

Line 233. We added the following. “It seemed possible that the 100 nm diameter size of our DEC1-AmB-LLs limited penetration and binding to cell wall beta-glucans. Therefore, we labeled hyphae with rhodamine-conjugated Dectin-1 protein. Dectin-1 is projected to have a diameter measured in tens of angstroms (Brown et al., 2008). We labeled fixed hyphae with DEC1-Rhod. The exopolysaccharide staining pattern was indistinguishable from that of DEC1-AmB-LLs (Supplemental Fig. SF2C).”

Line 335. We added the following. “Rhodamine tagged Dectin-1 protein bound with the same specificity to R. delemar’s exopolysaccharide. Hence, Dectin-1 appears to be responsible for DEC1-AmB-LL binding. Furthermore, it appears that the 100 nm size of DEC1-AmB-LLs did not significantly limit liposomal Dectin-1’s access to its cognate ligands.”

Line 344. We added the following address this concern. “AmB has several partially validated antifungal activities related to its affinity for ergosterol (Erg) in the fungal bilipid membrane, including opening ion channels in the membrane to cause lethal ion leakage and extracting Erg from the lipid bilayer to the membrane surface, which also compromises the membrane 31. Our results do not distinguish among the various mechanisms of AmB’s activity. Yet, our data robustly demonstrate that that Dectin-1 targeted DEC1-AmB-LLs were more efficiently associated with R. delemar’s exopolysaccharides and had greater antifungal activity than either AmB delivered in AmB-LLs or our protein-coated control BSA-AmB-LLs. Therefore, it does not appear that AmB itself plays a measurable role in the enhanced efficacy of targeted liposomes.”